# Field-tunable toroidal moment in a chiral-lattice magnet

Lei Ding [1,7], Xianghan Xu[2,7], Harald O. Jeschke [3], Xiaojian Bai[1], Erxi Feng[1], Admasu Solomon Alemayehu[2], Jaewook Kim[2], Fei-Ting Huang[2], Qiang Zhang [1], Xiaxin Ding [4], Neil Harrison [4], Vivien Zapf [4], Daniel Khomskii [5], Igor I. Mazin [6✉], Sang-Wook Cheong [2✉] & Huibo Cao [1✉]

Ferrotoroidal order, which represents a spontaneous arrangement of toroidal moments, has recently been found in a few linear magnetoelectric materials. However, tuning toroidal moments in these materials is challenging. Here, we report switching between ferritoroidal and ferrotoroidal phases by a small magnetic field, in a chiral triangular-lattice magnet $BaCoSiO_4$ with tri-spin vortices. Upon applying a magnetic field, we observe multi-stair metamagnetic transitions, characterized by equidistant steps in the net magnetic and toroidal moments. This highly unusual ferri-ferroic order appears to come as a result of an unusual hierarchy of frustrated isotropic exchange couplings revealed by first principle calculations, and the antisymmetric exchange interactions driven by the structural chirality. In contrast to the previously known toroidal materials identified via a linear magnetoelectric effect, $BaCoSiO_4$ is a qualitatively new multiferroic with an unusual coupling between several different orders, and opens up new avenues for realizing easily tunable toroidal orders.

[1] Oak Ridge National Laboratory, Neutron Scattering Division, Oak Ridge, TN, USA. [2] Rutgers Center for Emergent Materials and Department of Physics and Astronomy, Piscataway, NJ, USA. [3] Research Institute for Interdisciplinary Science, Okayama University, Okayama, Japan. [4] Los Alamos National Laboratory, Los Alamos, New Mexico, USA. [5] II. Physikalisches Institut, Universität zu Köln, Köln, Germany. [6] Department of Physics and Astronomy, George Mason University, Fairfax, VA, USA. [7] These authors contributed equally: Lei Ding, Xianghan Xu. ✉email: imazin2@gmu.edu; sangc@physics.rutgers.edu; caoh@ornl.gov

In localized spin systems, a toroidal moment, which violates space inversion and time-reversal symmetry, can be generated by a head-to-tail arrangement of magnetic moments[1,2]. Such a magnetic vortex can have two senses, corresponding to the opposite directions of the toroidal moment. Ferrotoroidicity, that is, the uniform arrangement of toroidal moments has been actively discussed as the fourth primary ferroic order, in addition to ferromagnetism, ferroelectricity, and ferroelasticity[2–9]. Ferrotoroidal order is thus of great fundamental and technological interest in condensed matter physics and spintronics[3,6,10,11]. From the symmetry-allowed terms in the free energy expression, toroidal moments should lead to antisymmetric components in the linear magnetoelectric effect. Indeed, discovered so far ferrotoroidal systems have been limited to a couple of linear magnetoelectric materials[5,12,13].

Ferrotoroidal phases in linear magnetoelectric materials have been studied using either polarized neutron analysis or second-harmonic generation technique[5,12,13]. The pioneering work on LiCoPO₄ using the latter method has revealed the presence of toroidal domains[5]. However, by the very nature of the toroidal ordering in this compound, a hysteretic poling of ferrotoroidic domains was only realized by crossed magnetic and electric fields[14]. It is worth noting that the observed signal in LiCoPO₄ results from a staggered arrangement of toroidal moments, i.e., it is a ferri-(not ferro)toroidal order, related to the collinear magnetic structure with bi-spin vortices[15]. Another case is pyroxene LiFeSi₂O₆, which also shows a finite off-diagonal magnetoelectric effect[13]. The control of toroidal domains in this compound was only possible by applying crossed magnetic and electric fields, as revealed by the polarized neutron analysis. This appears to be a common problem in these toroidal materials, which require simultaneous application of crossed magnetic and electric field, and is a consequence of symmetry constraints[12–14]. This restriction is directly related to their linear magnetoelectric effect, rendering these materials curious, but impractical.

A natural question to ask is whether it is possible at all to control toroidal moments using a single field. Our work presented below answers the question affirmatively, and clarifies the underpinning physical picture. The direct coupling between ferromagnetism and ferroelectricity in such chiral multiferroic materials as BaCoSiO₄ means that toroidal moments can be readily controlled using a single field, in a natural environment for the simultaneous existence of multi-dipole orders.

In a chiral structure, an object subjected to spatial inversion cannot be superimposed upon itself by any combination of rotations and translations[16,17]. Owing to this special symmetry, chirality has been found to be instrumental in stabilizing unusual magnetic orders such as multiferroicity[18,19], skyrmionic order[20–22], helicity[23,24], and chiral magnetic soliton lattice[25]. Chirality combined with the magnetic frustration characteristic of antiferromagnetic interactions in an equilateral triangle tends to generate a 120° vortex-like configuration[26,27], as shown in Fig. 1, giving rise to a nonzero toroidal moment breaking both the spatial inversion and time-reversal symmetry[26–28]. Depending on the sense of the in-plane spin rotations, the toroidal moment is either positive (+) or negative (−). Manipulating toroidal moments directly by a magnetic field becomes possible in such a chiral magnetic vortex, if an out-of-plane spin component is present and coupled with the in-plane spin texture through Dzyaloshinskii–Moriya (DM) interactions[29,30].

Following this strategy, in this work, we find a ferritoroidal order in a unique vortex-like spin configuration in the chiral magnet BaCoSiO₄. By applying a small magnetic field, the toroidal moments are uniformly aligned, thereby leading to a ferri- to ferrotoroidal transition. This toroidal transition, as well as the simultaneously scalar ferri- to ferrochiral transition, is fully

explained within a magnetic Hamiltonian accounting for the magnetic frustration and antisymmetric DM interactions. A key property of this Hamiltonian, as derived from first-principles calculations, is a rather special and not immediately obvious hierarchy of Heisenberg exchange parameters, which does not correlate with the length of the corresponding Co–Co bonds.

## Results

**Crystal structure and multi-stair metamagnetic transitions**. The stuffed tridymite BaCoSiO₄ crystallizes in the polar space group $P6_3$. Its crystal structure is therefore chiral and adopts only one enantiomorph[31]. Co atoms are tetrahedrally coordinated by oxygen with a large off-center distortion and the nearest Co atoms form spin trimers in the $ab$ plane (Fig. S1). In this structure, magnetic interactions between $Co^{2+}$ ions are expected to be small due to long and indirect exchange paths through adjacent SiO₄ tetrahedra. The temperature-dependent magnetic susceptibility shows an anomaly at $T_N \sim 3.2$ K (Fig. S5), reflecting a long-range antiferromagnetic magnetic order as discussed in the following. The Curie–Weiss law describes well the high-temperature ($150 \lesssim T \lesssim 300$ K) magnetic susceptibility, with negative Weiss temperatures $\theta_{CW}^{ab} = -10(2)$ K for $\mathbf{H} \| ab$ and $\theta_{CW}^c = -26.2(4)$ K for $\mathbf{H} \| c$. The fitted Curie constants correspond to effective moments $\mu_{eff}^{ab} = 4.6(4)\mu_B$ and $\mu_{eff}^c = 4.4(2)\mu_B$, consistent with the high-spin state of the $Co^{2+}$ cation with $S = 3/2$ and a nearly isotropic g-tensor of ≈2.3 (note that this deviates considerably from the nonrelativistic value $g = 2$, indicating sizeable spin–orbit effects). As shown in Fig. 2a, the magnetization hysteresis loop of BaCoSiO₄ at 2 K exhibits a sequence of metamagnetic transitions. Starting with a zero-field cooled sample, the first transition to $\sim0.1\mu_B$ is observed at low fields ($\leq150$ Oe) for $\mathbf{H} \| c$, stemming from an alignment of weak ferromagnetic domains. After a slow and linear ramp, a second transition to $\sim0.4\mu_B$ occurs at a critical field $\mu_0 H_C \sim 1.2$ T. This corresponds to a spin flip in one of the ferrotoroidal sublattices, as we will elaborate further below. Similar transitions occur with small hysteresis loops for reversed fields. Using a pulsed magnetic field, we measured the magnetization up to 60 T along different crystallographic directions (Fig. 2a, inset). A much higher field is needed to reach saturation with the magnetic field applied along the c-axis, which implies the presence of a considerable easy-plane anisotropy, formally not expected for $Co^{2+}$ in a tetrahedral environment, but consistent with $g > 2$. The slope changes around 4 and 7 T before the saturation suggest additional transitions of a completely different nature (Fig. S5).

**Spin vortex revealed by neutron diffraction**. To further characterize the magnetic ground state, we have measured the powder neutron diffraction of BaCoSiO₄ in a zero magnetic field. The diffraction pattern at 1.8 K shows a set of satellite reflections that can be indexed with a propagation vector $\mathbf{k} = (1/3, 1/3, 0)$ with respect to the crystallographic unit cell. The thermal evolution of the reflection (2/3 2/3 0) confirms the magnetic origin of the satellite reflections (Fig. 2b, inset). A power-law fit of the integrated intensity as a function of temperature gives a critical exponent $\beta = 0.37(1)$ and $T_N = 3.24(1)$ K, in agreement with the magnetization results. The symmetry analysis[32] and Rietveld refinement[33] based on the neutron data yield a complex magnetic structure where the in-plane components of magnetic moments form a vortex-like configuration (Fig. 2d). The structure is consistent with the magnetic space group $P6_3$ (No. 173.129) with a $\sqrt{3} \times \sqrt{3}$ magnetic supercell. To test the reliability of the refined in-plane spin orientation, we evaluate the profile factor $R_p$ of the fit as a function of uniform rotation ($\phi$) of spins within the $ab$ plane. Figure 2b reveals clear minima at $\phi \sim 0°$ (the structure in

| Chiral configuration | Left | | Right | |
|---|---|---|---|---|
| | | mirror | | |
| | 2-fold rotation | | 2-fold rotation | |
| Toroidal moment, **t** | − | + | + | − |
| Vector chirality, $\epsilon$ | + | + | + | + |
| Scalar chirality, $\kappa$ | + | − | + | − |

**Fig. 1 Toroidal moment and magnetic chirality of spin vortex configurations with 3-fold rotation symmetry.** Four chiral noncoplanar structures are divided into groups of two left-handed ones and two right-handed ones. Those within each group are connected by 2-fold rotations and those between two groups by a mirror operation. The spins (axial vector) are represented as arrows. The dashed line with an ellipsoid indicates one of the three 2-fold axes. Three relevant physical quantities are defined with spins {$\mathbf{S}_i = 1, 2, 3$} numbered anticlockwise: toroidal moment $\mathbf{t} = \sum_i \mathbf{r}_i \times \mathbf{S}_i$, where $\mathbf{r}_i$ is the vector from the center of the triangle to spin $\mathbf{S}_i$; vector chirality $\epsilon = \mathbf{S}_1 \times \mathbf{S}_2 + \mathbf{S}_2 \times \mathbf{S}_3 + \mathbf{S}_3 \times \mathbf{S}_1$; scalar chirality $\kappa = (\mathbf{S}_1 \times \mathbf{S}_2) \cdot \mathbf{S}_3$. Green symbols + and − for a toroidal moment and vector chirality denote the direction of these quantities with respect to the net magnetic moment, + for parallel and − for antiparallel. The magnetic vector chirality characterizes the sense of spin rotation along an oriented loop (or line), while the toroidal moment is associated with that around a center. Scalar spin chirality is a measure of non-coplanarity that does not necessarily have a sense of rotation. In the current example, toroidal moment and scalar chirality have a one-to-one correspondence to the net magnetic moment for a given handedness, since all of them are odd under time reversal. In a crystalline material with these chiral spin trimers as basic units, a ferro alignment of net magnetic moments of trimers will lead to a ferro ordering of toroidal moments and scalar chirality.

Fig. 2d) and $\phi \sim 120°$, indicating the global orientation of the spin structure with respect to the lattice is strongly constrained. The bulk magnetization data imply the existence of a weak ferromagnetic canting along the $c$-axis, which is allowed by the magnetic space group symmetry; however, it is too small to be unequivocally determined from current neutron data. A satisfactory fit can be achieved by setting canting angles to zero, yielding an ordered moment $m_{Co}(0\,T) = 2.71(5)\,\mu_B$ at 1.8 K and $\sim 3.67\,\mu_B$ at zero temperature from extrapolating the power-law fitting (Fig. S4). To appreciate the toroidal nature of the magnetic ground in BaCoSiO$_4$, it is instructive to decompose the structure into three interpenetrating sublattices (red, cyan, and blue) (Fig. 2d), each of which is a network of trimers (up- and down-triangles). Spins on every trimer form a 120° configuration that resembles a vortex and generates a nonzero toroidal moment. All trimers belonging to a single sublattice have an identical toroidal moment, giving rise to three ferrotoroidal sublattices. In zero field, two of them (red and blue) have the same total moment **t**, while the remaining one (cyan) has the opposite moment, leading to a net moment of −1**t** or +1**t** within a macroscopic magnetic domain. We dub this structure ferritoroidal.

The field dependence of the magnetic ground state in BaCoSiO$_4$ is investigated using single-crystal neutron diffraction. Figure 2c shows the integrated intensity of the magnetic reflection (2/3 2/3 0) and the nuclear reflection (−1 1 0) as a function of the magnetic field along the $c$-axis at 1.5 K. The former is increasingly suppressed by fields and eventually disappears at $\mu_0 H_C \sim 1.2$ T, while the latter gains significant extra intensity, indicating a field-induced transition to a $\mathbf{k} = 0$ magnetic structure. The refined $\mathbf{k} = 0$ magnetic structure has the same magnetic space group symmetry as the zero-field structure. The key difference is that the sublattice (cyan) with the opposite toroidal moment is flipped by 180° in the field, yielding a uniformly aligned toroidal moment for all three sublattices with a total toroidal moment +3**t** (Fig. 2e), termed ferrotoroidal. This remarkable field-induced ferri- to ferrotoroidal transition directly correlates with the metamagnetic

transition observed in the bulk magnetization measurements at the same critical field.

**Density functional theory calculations.** While the magnetic configuration, at first glance, seems nearly incomprehensibly complicated, it actually has a straightforward microscopic explanation. To show that, we first calculated the isotropic Heisenberg magnetic interactions using the density functional theory (DFT). Inspecting the Co $t_{2g}$ bands crossing the Fermi level, we found that the five shortest Co–Co bonds, with $d_{Co-Co}$ between 5.11 and 5.41 Å, all have hopping integrals of roughly the same order (Fig. S6), so they should be included in the minimal model (Fig. 3a). Next, we performed total energy calculations for selected spin configurations and determined exchange parameters $J$ by fitting the results to the mean-field energies of a Heisenberg model. Figure 3b shows the fitted model parameters as a function of onsite interaction $U$, using the room-temperature crystal structure as input in DFT calculations. All five exchange couplings are antiferromagnetic. Most interestingly, despite all bond lengths being similar, two interactions stand out as dominant, independent of $U$: the intralayer coupling $J_t$ and the interlayer coupling $J_z$. The Co atoms connected by these two bonds form the three interpenetrating sublattices shown in Fig. 2d. Within each sublattice, we expect that frustrated $J_t$ interactions impose 120° spin configurations on trimers that are antiferromagnetically coupled by $J_z$. This explains the major part of the experimentally determined magnetic structure. In addition, relativistic DFT calculations indicate a strong easy-plane single-ion anisotropy (~2 K, comparable to the dominant exchange interactions). This is nontrivial, since Co$^{2+}$ in a tetrahedral environment features a full $e_g$ shell and half-filled $t_{2g}$, and formally is not supposed to have a sizeable orbital moment. The key is that the CoO$_4$ tetrahedra are considerably distorted and moreover Co$^{2+}$ is strongly off-centered (Fig. 3a), so that the $e_g$ and $t_{2g}$ orbitals are actually mixed. This is also consistent with the pulsed-field magnetization measurements.

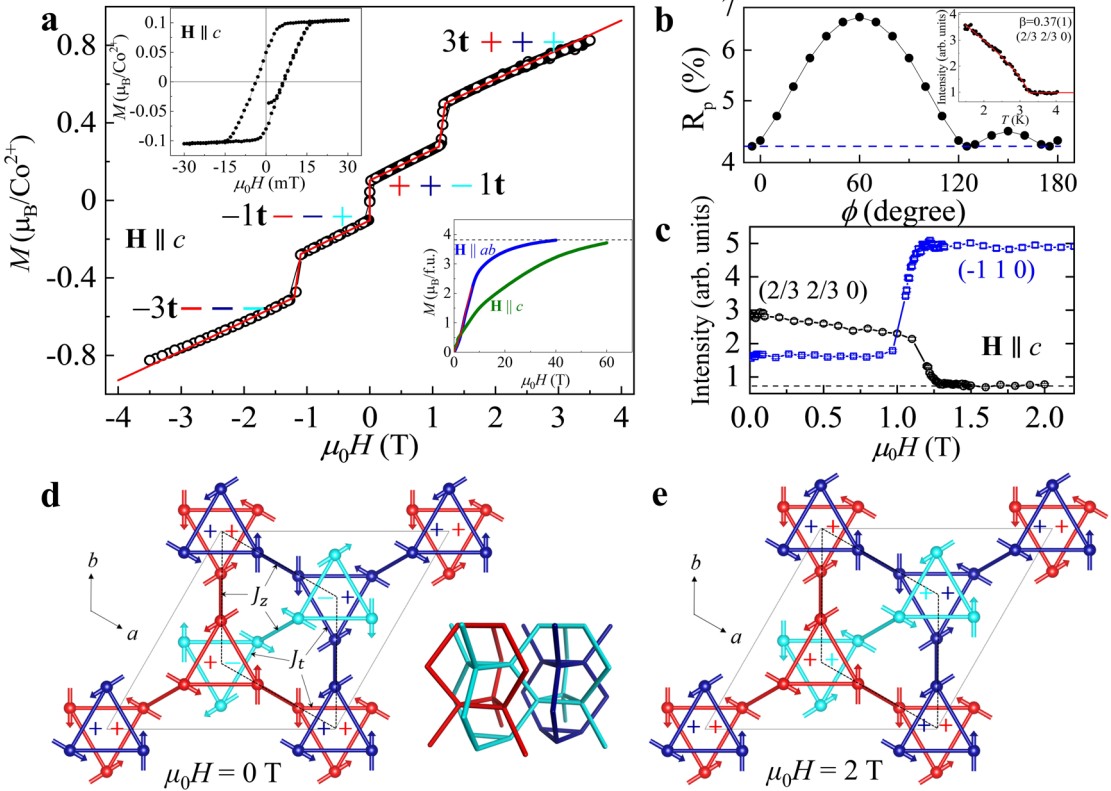

**Fig. 2 Zero-field magnetic structure and field-induced ferri- to ferrotoroidal transition in BaCoSiO₄. a** The isothermal bulk magnetization data (open circles) measured at 2 K with the field parallel to the $c$-axis and the theoretical calculations (red line) from the full spin Hamiltonian, showing a good agreement between data and calculations. The magnetization up to 60 T in a pulsed magnetic field at 1.5 K is shown in the lower inset, and the magnetization hysteresis at low fields in the upper inset. See caption of panel (**d**) for the meaning of symbols + and −. **b** The refined agreement factor for the powder neutron diffraction data as a function of uniform rotation in the $ab$ plane. The dashed line marks the best refinement. Inset shows the integrated intensity of the magnetic reflection (2/3 2/3 0) as a function of temperature with an order parameter fit $\sim (1 - T/T_N)^{2\beta}$ (solid line), where $\beta$ is the critical exponent. The error bars are used to show the standard deviation given by the square root of the number of neutron counts. **c** The integrated intensities of reflection (2/3 2/3 0) and (−1 1 0) as a function of the magnetic field with **H**‖$c$ at 1.5 K. **d** Zero-field magnetic structure of BaCoSiO₄ in a $\sqrt{3} \times \sqrt{3}$ supercell solved from powder neutron diffraction data, showing three interpenetrating ferritoroidal sublattices (red, blue, and cyan) formed by the dominant exchange interactions $J_t$ (intralayer) and $J_z$ (interlayer). The direction of the toroidal moment for each sublattice is denoted + if it is parallel to the $c$-axis and − if antiparallel. The red and blue sublattices have the same toroidal moment, while the cyan has the opposite moment, leading to a ferritoroidal state with a total moment +1**t**. The primitive crystallographic unit cell is indicated by the dotted lines. **e** Magnetic structure of BaCoSiO₄ at 2 T solved from single-crystal neutron diffraction data. All spins on the cyan sublattice are reversed, leading to a ferrotoroidal state with a total toroidal moment +3**t**. Triangles in panels (**d**, **e**) lie in two adjacent layers, which are bridged by the interlayer interaction $J_z$.

Of the three sublattices in Fig. 2d, one (cyan) has its toroidal moment opposite to the others. This results from the subleading exchange interactions $\{J'_t, J''_t, J_c\}$ (Fig. 3a). The first two connect the trimers within the $ab$ plane, while the last one connects those along the $c$-axis. A close inspection of the lattice connectivity reveals that all three subleading interactions with the help of $J_t$ form various triangular units. The total energy associated with subleading interactions is the lowest if spins on all these triangular units have 120° arrangements (neglecting the weak ferromagnetic canting). Yet, this condition cannot be satisfied. Figure 3d colors all "frustrated" triangles that do not have 120° configurations in a ferri- and ferrotoroidal state. We see clear switching of colored triangles from one state to another, indicating a direct competition between the intralayer $J'_t$ and $J''_t$ couplings and the interlayer $J_c$ couplings. In BaCoSiO₄, this competition favors a ferritoroidal state by a small margin of energy in zero field.

**Antisymmetric DM interactions.** However, this Heisenberg model, with (or without) the single-ion anisotropy, is insufficient

in explaining the weak ferromagnetic canting and the spin–space anisotropy evidenced from our experimental data. To this end, we introduce the antisymmetric DM interactions[28–30] within the structural trimers (Fig. 3a). All three components of a DM vector on a nearest-neighbor bond are allowed due to the lack of symmetry constraints. It is rather difficult to calculate the DMI from the first principles, so for the moment we are assuming that all three components are present. The DM vectors on different bonds of the trimer are related by the 3-fold rotations. Assuming a uniform canting along **c** and 120° configurations in the $ab$ plane for a trimer, the out-of-plane DM component $D_z\hat{\mathbf{z}}$ contributes $-\frac{3\sqrt{3}}{2}|D_z|S_{xy}^2$ in energy, where $S_{xy}$ is the length of in-plane spin component. Therefore, this term always favors coplanar spin configurations instead of canting. Taking into account that the 120° configuration has a 3-fold vortex symmetry, we find that the energy associated with the in-plane DM component, $\mathbf{D}_{xy}$, is $3\sqrt{3}(\mathbf{D}_{xy} \cdot \mathbf{S}_{xy})(\mathbf{S}_z \cdot \hat{\mathbf{z}})$, where $\mathbf{S}_z$ is the out-of-plane spin component. There are two important ramifications. First, the two components of spins are locked together, namely if $\mathbf{S}_z$ flips, so does $\mathbf{S}_{xy}$, to keep the DM energy gain. This is actually the essential physical factor that couples the net component of magnetization

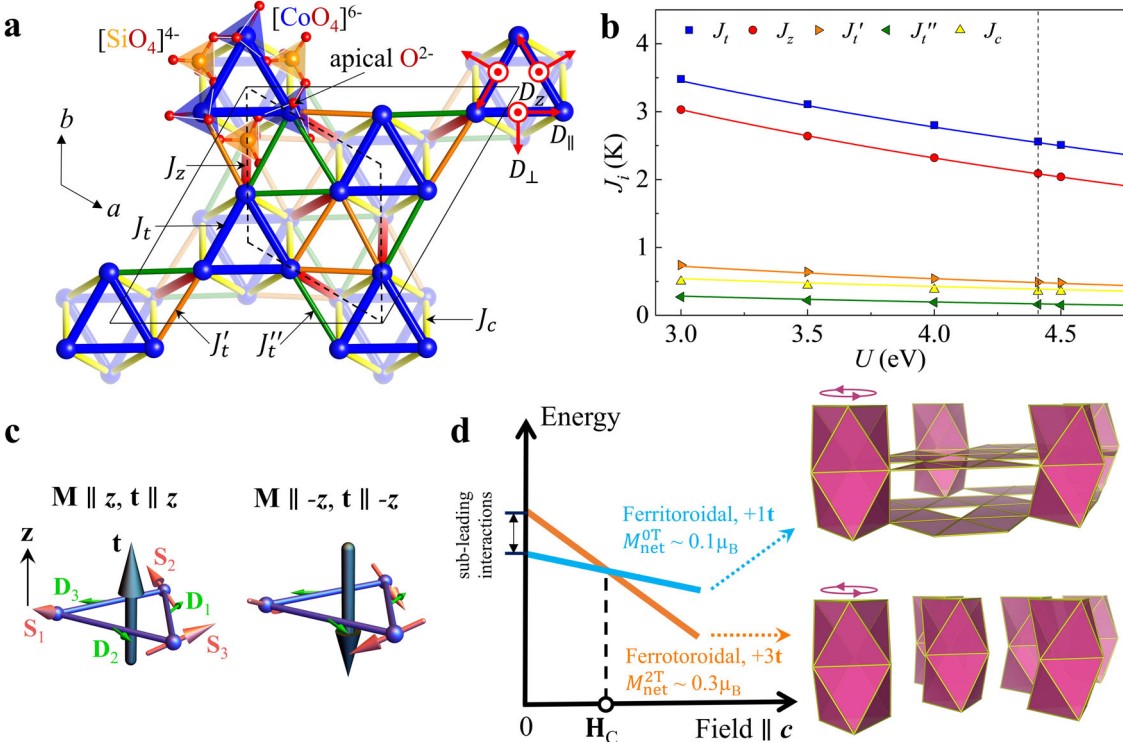

**Fig. 3 Microscopic magnetic model and the underlying mechanism for the ferritoroidal to ferrotoroidal transition. a** Magnetic exchange pathways of BaCoSiO$_4$, showing three intralayer couplings $\{J_t, J_t', J_t''\}$ and two interlayer couplings $\{J_z, J_c\}$. Three components of a DM vector on the nearest-neighbor $J_t$ bonds are indicated by red arrows in a local reference frame. **b** Density functional theory calculation of exchange interaction strengths as a function of onsite interaction $U$. The dashed line marks the set of couplings with $U = 4.41$ eV that matches the Weiss temperature from the magnetic susceptibility (Table S2). **c** Minimal energy configurations for three spins $\{S_i, i = 1, 2, 3\}$ on a triangle with an antiferromagnetic Heisenberg interaction and an in-plane DM interaction. The DM vectors $\{D_i, i = 1, 2, 3\}$ (green arrows) are related by 3-fold rotation symmetry and have the same sense of rotation as the tilting of apical oxygens shown in panel (**a**). Each vector makes an ~30° angle with the bond, see main text for details. For this set of DM vectors, the resulting spin structure (pink arrows) generates a toroidal moment $t$ (black arrows) that is always parallel to the magnetization $M$. **d** Energy balance between the ferri- to ferrotoroidal state in magnetic fields. The "frustrated" triangular units that do not have 120° configurations (and cost more energy) are highlighted in pink for both states. The ferrotoroidal state has less colored triangles in the $ab$ plane; therefore, it is energetically favored by the interactions $\{J_t', J_t''\}$, similarly the ferritoroidal state is favored by the interaction $J_c$. Competition between these subleading interactions results in the ferritoroidal structure with a lower energy in zero field. The transition to the ferrotoroidal state occurs when the energy difference is compensated by the Zeeman energy in magnetic fields.

$M_z$ to the toroidal moment and allows to control toroidal moments by altering $M_z$, e.g., by external fields. Second, to minimize the DM energy for a fixed spin canting, $S_{xy}$ has to be antiparallel to $D_{xy}$ if $S_z \parallel \hat{z}$ or parallel to $D_{xy}$ if $S_z \parallel -\hat{z}$, which means that we can read off the direction of the DM vector directly from the experimental spin structure, assuming $D_z = 0$. In Fig. 3c, we show by green arrows the total DM vector for each bond, which is the vector sum of $D_\perp$ and $D_\parallel$ components and makes a ~30° angle with the bond, as determined from the magnetic structure. For each sublattice connected by $J_t$ and $J_z$ bonds, the DM interaction creates a uniform $c$-axis canting and corresponding in-plane toroidal spin texture. However, different sublattices are independent of each other, hence ferri- and ferrotoroidal states would have had the same energy, if not for the subleading Heisenberg interactions $\{J_t', J_t'', J_c\}$.

## Discussion

We now fully understand the metamagnetic ferrotoroidal transition. Indeed, because of the DMI-induced ferromagnetic canting, the small $\{J_t', J_t'', J_c\}$-driven energy gain associated with the ferritoroidal arrangement competes directly with the Zeeman interaction favoring the ferrotoroidal phase. This leads to the spin flip (also a toroidal flip) manifested through the sudden increase of magnetization along the $c$-axis by a factor of three at the

metamagnetic transition (Fig. 2e). A direct numerical calculation of magnetization using the complete model with all interactions discussed thus far is given in Fig. 2a and shows good agreement with the data. See Supplementary information and "Method" section for more details and model parameters.

At a fundamental level, the physics emerging in BaCoSiO$_4$ originates from its chiral crystal structure. For a single triangle in three dimensions, there is a set of three mirror planes perpendicular to the triangle and one extra mirror plane containing the triangle. The former allows both $D_z$ and $D_\perp$ components of the DM interaction, while the latter allows only $D_z$. When $D_\perp$ is absent, $D_z$ with the correct sign could create a vortex configuration; however, the ferromagnetic canting is unfavored in this case. Thus, the intimate coupling between magnetization and toroidal moment is lost. In BaCoSiO$_4$, the triangles are decorated by distorted CoO$_4$ tetrahedra that break all the mirror planes while still preserving the 3-fold symmetry. The $D_\parallel$ term is then allowed and plays an essential role in generating chiral vortex structures wherein the $c$-axis canting is coupled with the sense of the spin rotation in the $ab$ plane. A direct control of toroidal moments using only magnetic fields is therefore possible through controlling the bulk magnetization by a uniform magnetic field, instead of using a conjugate field such as the curl of magnetic fields. The same arguments show that at this transition the scalar chirality $\kappa$ (Fig. 1), existing in Co triangles, also changes from the ferrichiral

to ferrochiral state, similar to the three-sublattice description for the toroidal transition. The ferrochiral state with a noncoplanar spin configuration acquires a considerable Berry curvature, which can lead to a variety of exotic physical phenomena such as topological magnon excitations[34,35].

In summary, we studied a rare chiral triangular-lattice magnet BaCoSiO$_4$ through bulk magnetization measurements and neutron diffraction experiments. We uncovered a novel vortex-like spin texture and a magnetic field-induced ferri- to ferrotoroidal transition for the first time. Combining ab initio DFT calculations and theoretical modeling, we have derived the microscopic energy balance and were able to explain quantitatively the complex magnetic structure and the field-induced metamagnetic/toroidal phase transition, neither of which had been observed before in any compound. Our work shows that BaCoSiO$_4$ is an excellent platform to study field-tunable toroidal moments and to explore their interplay with the structural and magnetic chirality. Further studies on the magnetoelectric effects and dynamical responses of toroidal spin textures are liable to bring up further new physics and potential applications.

## Methods

**Sample preparation and characterization.** A powder sample of BaCoSiO$_4$ was prepared by the direct solid-state reaction from stoichiometric mixtures of BaCO$_3$, Co$_3$O$_4$, and SiO$_2$ powders all from Alfa Aesar (99.99%). The mixture was calcined at 900 °C in the air for 12 h and then re-grounded, pelletized, and heated at 1200 °C for 20 h and at 1250 °C for 15 h with intermediate grindings to ensure a total reaction. Finally, the sample was rapidly quenched from 1250 °C to room temperature to avoid the decomposition at intermediate temperature. The resulting powder sample is fine and bright blue in color. Large single crystals were grown using a laser-diode heated floating zone technique. The optimal growth conditions were growth speed of 2–4 mm/h, atmospheric air flow of 0.1 L/min and counter-rotation of the feed and seed rods at 15 and 30 r.p.m., respectively. Single-crystal x-ray diffraction data were collected at 95 K using a Rigaku XtaLAB PRO diffractometer with the graphite monochromated Mo $K_\alpha$ radiation ($\lambda = 0.71073$ Å) equipped with a HyPix-6000HE detector and an Oxford N-HeliX cryocooler. Peak indexing and integration were done using the Rigaku Oxford Diffraction CrysAlisPro software. An empirical absorption correction was applied using the SCALE3 ABSPACK algorithm as implemented in CrysAlisPro. Structure refinement was done using the FullProf suite[33].

**Magnetization measurement.** Temperature dependence of magnetization $M(T)$ was measured under a field of 0.1 T in a commercial magnetic property measurement system (MPMS-XL7, Quantum Design). Magnetic hysteresis loops with the field along the a- and c-axis were measured at 2 K using the same setup. A 2.6 mg piece oriented BaCoSiO$_4$ single crystal was glued on a 3 mm × 3 mm piece of weighing paper by GE varnish first, and then they were mechanically fixed on a straw attaching to the MPMS sample rod. This mounting setup was tested without placing any sample and it shows a diamagnetic background signal around $-10^{-6}$ emu at ground temperature, whereas the BaCoSiO$_4$ sample shows $>10^{-3}$ emu magnetization. Thus, a significant influence from sample mounting can be ruled out. The core diamagnetism has a typical magnitude of $10^{-6}$ emu/mole, which is also negligible, compared with the observed susceptibility signal magnitude. Magnetization up to 60 T was measured by an induction magnetometry technique[36] using a capacitor-bank-driven pulsed magnet at the National High Magnetic Field Laboratory pulsed-field facility at Los Alamos. The pulsed-field magnetization values are calibrated against measurements in a 7 T direct current magnet (MPMS-XL7, Quantum Design).

**Neutron diffraction.** Neutron powder diffraction experiments were performed on the time-of-flight powder diffractometer POWGEN at the Spallation Neutron Source at Oak Ridge National Laboratory (ORNL). A powder sample of ~2 g was loaded in a vanadium cylinder can and measured in the temperature range of 1.8–10 K with neutron wavelength band centered at $\lambda = 1.5$ and 2.665 Å, covering the d-space range 0.5–9.0 and 1.1–15.4 Å, respectively. Single-crystal neutron diffraction experiments were carried out on the single-crystal neutron diffractometer HB-3A DEMAND equipped with a 2D detector at the High Flux Isotope Reactor, ORNL. The measurement used the neutron wavelength of 1.553 Å selected by a bent perfect Si-220 monochromator[37,38]. The single crystal (~0.2 g) was mounted in a vertical field superconducting cryomagnet with a magnetic field up to 5.5 T and measured over the temperature range of 1.5–10 K with a magnetic field applied along the c-axis. The data refinements were performed by the FullProf suite[33].

**Spin-polarized DFT calculations.** Electronic structure calculations were performed using the full potential local orbital basis[39] and generalized gradient approximation exchange and correlation functional[40]. The crystal structure from ref. [31] was used. We correct for the strong electronic correlations on Co 3d orbitals using a DFT+U method[41] with a fixed value of the Hund's rule coupling $J_H = 0.84$ eV as suggested in ref. [42]. The Heisenberg Hamiltonian parameters were determined by an energy mapping technique[43].

**Full spin model calculations.** The experimental magnetization data shown in Fig. 2a was modeled using a full spin Hamiltonian, including exchange interactions, single-ion anisotropy, and external fields,
$\mathcal{H} = \frac{1}{2}\sum_{ij} J_{ij} \mathbf{S}_i \cdot \mathbf{S}_j + A\sum_i (S_i^z)^2 - g\mu_B\mu_0\sum_i \mathbf{H} \cdot \mathbf{S}_i$, with $g = 2$ and $S = 3/2$. Starting with the Heisenberg interactions strengths produced by DFT calculations, by trial and error, we found the following set of parameters reasonably reproducing the experimental magnetization data, $J_t = 2.22$ K, $J_z = 1.87$ K, $J_t' = 0.21$ K, $J_t'' = 0.61$ K, $J_c = 0.48$ K, $D_\parallel = 1.71$ K, $D_\perp = D_\parallel/\sqrt{3}$, $D_z = 0$ K, and $A = 7.5$ K. Direct numerical minimization of the classical energy was performed for this model using a $\sqrt{3} \times \sqrt{3}$ supercell. The magnetization is obtained by averaging all spins of the lowest energy configuration at each field value, shown as the red line in Fig. 2a.

## Data availability

The data that support the findings of this work are included in the article and the Supplementary information file. All raw data in the current work are available from the corresponding authors on reasonable request. The X-ray crystallographic coordinates for structures reported in this study have been deposited at the Cambridge Crystallographic Data Center (CCDC), under deposition number CSD 2100492. These data can be obtained free of charge from The Cambridge Crystallographic Data Center via www.ccdc.ac.uk/data_request/cif.

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

## Acknowledgements

The work at Oak Ridge National Laboratory (ORNL) was supported by the US Department of Energy (DOE), Office of Science, Office of Basic Energy Sciences, Early Career Research Program Award KC0402020, under Contract No. DE-AC05-00OR22725. This research used resources at the High Flux Isotope Reactor and the Spallation Neutron Source, the DOE Office of Science User Facility operated by ORNL. The work at Rutgers University was supported by the DOE under Grant No. DOE: DE-FG02-07ER46382. The work of D.K. was funded by the Deutsche Forschungsgemeinschaft (DFG, German Research Foundation)—Project number 277146847—CRC 1238. I.I.M. acknowledges support from DOE under Grant No. DE-SC0021089. The National High Magnetic Field Laboratory is funded by the US National Science Foundation through Cooperative Grant No. DMR-1157490, the US DOE, and the State of Florida. N.H. acknowledges support from DOE BES project "Science of 100 tesla".

## Author contributions

S.-W.C. conceived the BaCoSiO₄ project. H.C., S.-W.C., and I.I.M. supervised this work. X.X. and A.S.A. grew the sample. X.X., J.K., F.H., X.D., N.H., V.Z., and S.-W.C. measured bulk magnetization data. L.D., E.F., Q.Z., and H.C. performed neutron diffraction experiments and data analysis. H.O.J., X.B., D.K., and I.I.M. performed DFT calculations and theoretical modeling. L.D., X.B., I.I.M., S.-W.C., and H.C. wrote the paper with comments from all the authors.

## Competing interests

The authors declare no competing interests.
