## [Peer Review File · Nature Communications]

Reviewers' Comments:

Reviewer #1:

Remarks to the Author:

Having refereed this paper in its previous form, I am satisfied by the authors' response and revisions and recommend publication in *Nature Communications*. My previous statements on the importance and relevance of the paper stand. I believe this material and the data and interpretation that the authors have provided are extremely interesting and of very high potential impact.

The additional details that have been included in the manuscript and supplementary information are now sufficient for readers to judge the data quality and interpretation objectively. I find that the authors' interpretation is reasonable, but with such complex physics at play such results will always be liable to reinterpretation due to e.g. advances in theory. Including full details of sample characterisation and analytical and experimental procedures allow the data and interpretation to stand the test of time in a fair manner. I believe this manuscript is suitable for *Nature Communications* and represents a significant advance in the field of complex ferroic orders.

Reviewer #2:

Remarks to the Author:

After reviewing the author's responses to my original inquiries and reading the revised manuscript, my original concerns have been addressed. I believe the manuscript, presentation, and discussion have been greatly improved and are suitable for publication in *Nature Communications*.

Point-by-point response to the reviewers' comments:

Reviewer #1 (Remarks to the Author):

Having refereed this paper in its previous form, I am satisfied by the authors' response and revisions and recommend publication in Nature Communications. My previous statements on the importance and relevance of the paper stand. I believe this material and the data and interpretation that the authors have provided are extremely interesting and of very high potential impact.

The additional details that have been included in the manuscript and supplementary information are now sufficient for readers to judge the data quality and interpretation objectively. I find that the authors' interpretation is reasonable, but with such complex physics at play such results will always be liable to reinterpretation due to e.g. advances in theory. Including full details of sample characterisation and analytical and experimental procedures allow the data and interpretation to stand the test of time in a fair manner. I believe this manuscript is suitable for Nature Communications and represents a significant advance in the field of complex ferroic orders.

Reply:

We appreciate the referee's recommendation and positive comments about the implication of our work. We thank the referee's constructive suggestions in the first round about adding more technical or experimental details which, we finally realized, are important for general readers.

Reviewer #2 (Remarks to the Author):

After reviewing the author's responses to my original inquiries and reading the revised manuscript, my original concerns have been addressed. I believe the manuscript, presentation, and discussion have been greatly improved and are suitable for publication in Nature Communications.

Reply:

We thank the referee's recommendation for publication.